# No Long-Term Mucosal Lesions in the Esophagus but More Gastric Mucosal Lesions after Sleeve Gastrectomy in Obese Rats

**DOI:** 10.3390/jcm12051848

**Published:** 2023-02-25

**Authors:** Muriel Coupaye, Lara Ribeiro-Parenti, Clément Baratte, Muriel Hourseau, Alexandra Willemetz, Henri Duboc, Séverine Ledoux, André Bado, Anne Couvelard, Maude Le Gall

**Affiliations:** 1UMRS 1149 Centre de Recherche sur l’Inflammation, Université Paris Cité, Inserm, 75018 Paris, France; 2Centre Intégré Nord Francilien de Prise en Charge de l’Obésité (CINFO), Assistance Publique-Hôpitaux de Paris, Service des Explorations Fonctionnelles, Hôpital Louis-Mourier, 92700 Colombes, France; 3Centre Intégré Nord Francilien de Prise en Charge de l’Obésité (CINFO), Assistance Publique-Hôpitaux de Paris, Service de Chirurgie Digestive, Hôpital Bichat-Claude-Bernard, 75018 Paris, France; 4Assistance Publique-Hôpitaux de Paris, Service d’Anatomo-Pathologie, Hôpital Bichat-Claude-Bernard, 75018 Paris, France; 5Assistance Publique-Hôpitaux de Paris, Service de Gastroentérologie, Hôpital Louis-Mourier, 92700 Colombes, France

**Keywords:** rats, obesity, bariatric surgery, sleeve gastrectomy, esogastric mucosa, gastroesophageal reflux disease, Barrett’s esophagus

## Abstract

Sleeve gastrectomy (SG) often induces gastroesophageal reflux, with few and discordant long-term data on the risk of Barrett’s esophagus (BE) in operated patients. The aim of this study was to analyze the impact of SG on esogastric mucosa in a rat model at 24 weeks postoperatively, which corresponds to approximately 18 years in humans. After 3 months of a high-fat diet, obese male Wistar rats were subjected to SG (n = 7) or sham surgery (n = 9). Esophageal and gastric bile acid (BA) concentrations were measured at sacrifice, at 24 weeks postoperatively. Esophageal and gastric tissues were analyzed by routine histology. The esophageal mucosa of the SG rats (n = 6) was not significantly different in comparison to that of the sham rats (n = 8), with no esophagitis or BE. However, there was more antral and fundic foveolar hyperplasia in the mucosa of the residual stomach 24 weeks after SG than in the sham group (*p* < 0.001). Luminal esogastric BA concentrations did not differ between the two groups. In our study, SG induced gastric foveolar hyperplasia but no esophageal lesions at 24 weeks postoperatively in obese rats. Therefore, long-term endoscopic esophageal follow-up that is recommended in humans after SG to detect BE may also be useful for detecting gastric lesions.

## 1. Introduction

Bariatric surgery is the most effective treatment for severe obesity and its comorbidities [1]. Among bariatric procedures, sleeve gastrectomy (SG) has become the most commonly performed procedure worldwide since 2014 [2] because it induces fewer surgical complications and vitamin deficiencies than malabsorptive procedures [3].

SG reduces gastric volume by resecting the majority of the corpus of the stomach along the greater gastric curvature and constructing a tubular gastric pouch [4]. The main adverse effect of SG is gastroesophageal reflux disease (GERD) [5], with concerns about the development of Barrett’s esophagus (BE), a condition in which metaplastic columnar mucosa replaces the esophageal squamous mucosa damaged by GERD [6]. Given the 0.1–0.3% annual risk of developing esophageal adenocarcinoma in the presence of BE in the general population [7], early detection of BE with close endoscopic follow-up after SG is currently recommended [8,9]. However, the results concerning BE after SG are controversial, ranging from less than 2% [10,11] to approximately 20% [12,13] at 5 years postoperatively. The follow-up of the most recent endoscopic studies is only up to 10 years [14,15,16], because SG is a relatively new procedure worldwide. Only one paper has reported endoscopic findings at 15 years in 20 patients, finding BE in 13% of them [17]. This latter paper is also the only one to report findings of gastric biopsies during endoscopy. The authors found active gastritis in more than 70% of the 20 patients, without significant differences between patients with or without GERD [17].

Rats are recognized as a good model of esophageal carcinogenesis [18]. However, only one study has reported the consequences of SG on the esophageal mucosa of Wistar rats, and it found more severe esophagitis in SG rats compared to sham rats at 12 weeks postoperatively [19]. Moreover, only three studies have assessed the consequences of SG on the gastric mucosa of Wistar rats, with a maximum follow-up of 16 weeks postoperatively, and they reported contrasting results, from normal fundic mucosa [20] to gastric foveolar hyperplasia and cystic glandular dilatation [21,22].

Therefore, the aim of this study was to evaluate the consequences of SG on the esophageal and gastric mucosa in obese rats at 24 weeks postoperatively, which corresponds to approximately 18 years postoperatively in humans [23].

## 2. Materials and Methods

### 2.1. Ethics

All animal studies were conducted in compliance with EU directives for animal experimentation and were approved by the Ethical Committee of Paris North and the French Minister of Higher Education, Research and Innovation (APAFIS #8290). Male Wistar rats (6 weeks old) weighing 220–240 g were fed a high-fat diet (HFD) (Altromin C45, Genestil, Royaucourt, France) for 3 months.

### 2.2. Animal Surgery and Post-Surgery Procedures

Sixteen male Wistar rats (mean weight: 595.8 ± 86.6 g) were randomly assigned to sham surgery (n  =  9), or SG surgery (n  =  7). Surgical models have previously been described in detail [24]. All procedures were performed by the same surgeon. Briefly, for SG, the first step led to an 80% resection of the fundus of the stomach using one application of an ETS-Flex 35 mm staple gun (Ethicon, Issy les Moulineaux, France), leaving a thin gastric tube in continuity with the esophagus, as in humans. The antrum was kept in place with the help of a bougie 6 introduced through the forestomach, a non-glandular part of the stomach. The second step was the resection of the forestomach using one application of an ETS-Flex 35 mm staple gun (Ethicon, Issy les Moulineaux, France). For sham-operated rats, the stomach was pinched with an unarmed staple gun without sectioning. The SG and sham rats had free access to water during day 1 post-surgery, to a liquid diet on days 2 and 3 post-surgery, and to a normal diet on day 4 post-surgery. The rats were kept in individual cages throughout the experimental period. Body weight and food intake were recorded daily for 2 weeks, and then weekly.

The overall survival rate was 87%, with 2 deaths among the 16 rats occurring in the immediate postoperative period: one in the SG group (gastric leak) and one in the sham group (perioperative shock). The rats were sacrificed at 24 weeks postoperatively (6 SG and 8 shams) by exsanguination under deep sedation.

### 2.3. Luminal Bile Acid Concentrations

After euthanasia, the esophagi and stomachs were removed and flushed with 500 µL of phosphate-buffered saline and then stored at −80 °C until analyses. Bile acids were extracted by solid-phase extraction and analyzed using high-performance liquid chromatography tandem mass spectrometry, as previously described [25].

### 2.4. Histological Analyses

After euthanasia, the esophagus and stomach segments were fixed overnight in 10% neutral buffered formalin, paraffin-embedded, and sectioned at 4 μm longitudinally from esophagus to duodenum. The sections were then stained with hematoxylin–eosin–safran (HES).

The esophageal mucosa was examined for esophagitis, Barrett’s esophagus, dysplasia, and cancer. Esophagitis was defined by the association of basal cell hyperplasia and inflammatory cell infiltration. Esophageal hyperkeratosis (EHK) was defined as esophageal keratin height greater than the height of the esophageal mucosa in at least 3 locations. Esophageal hyperpapillomatosis (EHP) was defined as the presence of epithelial crests with a sinuous appearance of the esophageal epithelium and basal membrane. In the gastric mucosa, metaplasia, dysplasia, cancer, and foveolar hyperplasia (FH), a feature of reactive gastritis or gastropathy, were sought in the antrum and fundus. FH was defined as crypt hyperplasia. The slides were interpreted by two pathologists (AC and MH) who were blinded to the procedure.

### 2.5. Statistical Analyses

All values are expressed as means ± SEM. All comparisons used non-parametric tests: Mann–Whitney tests for quantitative variables and Fisher’s exact tests for qualitative variables. Statistical analyses were performed with GraphPad Prism version 9.1.2 (GraphPad Software, San Diego, CA, USA). A value of *p* < 0.05 was considered statistically significant.

## 3. Results

### 3.1. Evolution of Body Weight and Food Intake at 24 Weeks Postoperatively

Figure 1 shows the changes in percent preoperative weight and food intake up to 24 weeks postoperatively. Both SG and sham rats experienced maximal weight loss 15 days postoperatively (Figure 1A), with significantly greater weight loss in SG rats than in sham rats (−9.3% versus −4.4% of initial weight, respectively, *p* < 0.001). Then, weights increased in both groups but remained lower in SG rats compared with sham rats at 24 weeks postoperatively (+2.0% versus +9.0% of initial weight, respectively, *p* = 0.012).

Food intake became similar in both groups as soon as 15 days postoperatively (Figure 1B) and remained similar at 24 weeks postoperatively (27.8 ± 2.7 g/day for SG rats versus 26.6 ± 0.8 g/day for sham rats, *p* = 0.38).

### 3.2. Luminal Esogastric Bile Acid Content at 24 Weeks Postoperatively

No significant differences were found between SG and sham rats at 24 weeks postoperatively in total luminal BA concentrations (1323 ± 554 µmol/L for sham versus 1634 ± 269 µmol/L for SG, *p* = 0.23), primary BA concentrations (1035 ± 421 µmol/L for sham versus 1393 ± 230 µmol/L for SG, *p* = 0.23), and secondary BA concentrations (258 ± 122 µmol/L for sham versus 222 ± 51 µmol/L for SG, *p* = 0.49).

### 3.3. Histological Analyses of the Esophageal Mucosa at 24 Weeks Postoperatively

HES staining of the esophageal mucosa of both groups (Figure 2) was classified as healthy esophageal mucosa (Figure 2A), EHK (Figure 2B), or EHP (Figure 2C). EHK and EHP were not significantly different between sham and SG rats (*p* = 0.32, Figure 2D and *p* = 0.29, Figure 2E, respectively). No esophagitis or Barrett’s esophagus were observed.

### 3.4. Histological Analyses of the Gastric Mucosa at 24 Weeks Postoperatively

As previously published by our team [20], macroscopic examination of the residual stomachs after SG revealed that the antral and fundic surfaces were largely increased at 24 weeks postoperatively.

HES staining of the antral and fundic mucosa of both groups (Figure 3) was classified as healthy antral mucosa (Figure 3A), AFH (Figure 3B), healthy fundic mucosa (Figure 3C), or FFH (Figure 3D). AFH and FFH occurred significantly more often in the SG group compared with the sham group (*p* < 0.001, Figure 3E,F, respectively).

Antral intestinal metaplasia was found in one SG rat (Figure 4). No gastric dysplastic or cancerous lesions were observed.

## 4. Discussion

Due to the lack of long-term data after SG in humans, significant uncertainty remains regarding GERD-induced esogastric lesions after SG. Thus, we developed a rat model to explore the long-term consequences of SG on the esogastric mucosa. To our knowledge, this is the first study to report the consequences of SG on the esogastric mucosa in a rat model at 24 postoperative weeks. This model allowed us to show that SG induces more frequent gastric foveolar hyperplasia in HFD obese rats compared to sham rats, but we were unable to provide evidence of esophagitis or esophageal metaplasia after SG.

In our study, SG rats had significantly greater weight loss from 15 days postoperatively compared with sham rats, and then weights increased in both groups but remained lower in SG rats compared with sham rats at 24 weeks postoperatively. These results have been reported previously in Wistar rats at 5 weeks [20] and 16 weeks after SG [21]. Interestingly, food intake became similar from 15 days postoperatively and remained the same at 24 weeks postoperatively. These results have also been reported in rats at 5 weeks [20], 12 weeks [21], and 16 weeks after SG [21]. In humans, weight regain after SG is also often observed after 2 or 3 postoperative years, and it is always observed in studies with 10- or 15-years postoperative follow-up [14,15,16,17].

In our study, we found no esophagitis and no difference in esophageal lesions between SG rats and sham rats at 24 weeks postoperatively, and no Barrett’s esophagus was observed. Only one study has investigated esophageal injury after SG in Wistar rats [19]. The authors reported more severe esophagitis lesions at 12 weeks postoperatively in SG rats compared with sham rats. Furthermore, they did not report the presence of EHP in either group, although the presence of approximately 50% EHP was reported in Wistar sham rats at 16 [26] and 30 weeks postoperatively [27]. In this latter study, 30% EHP was also reported after one-anastomosis gastric bypass (OAGB) and 10% EHP after Roux-en-Y gastric bypass at 30 weeks postoperatively [27]. In contrast, a recent study of diabetic Sprague-Dawley rats reported more severe EHP in SG rats compared with sham rats at 12 weeks postoperatively [28]. A possible explanation for the difference between these studies could be the genetic background of the rats, the phenotypic differences of these models (diabetes or not), or the type of SG, as the forestomach (non-glandular part of the stomach) does not appear to be completely removed in the Altieri [19] and Wang [28] studies. Nevertheless, SG does not appear to have as much impact on the esophagus in rats as OAGB, with more than 50% of esophagitis previously described in male Wistar rats operated by OAGB with a long biliopancreatic limb [27].

To our knowledge, only one study has reported the results of gastric biopsies performed at 15 years postoperatively in 16 patients. The authors found “active gastritis” in 75% of the 16 patients, but no other type of gastric lesion was described [17]. Furthermore, the prevalence of gastric intestinal metaplasia in postoperative gastric biopsies has never been reported in humans, whereas this prevalence is estimated to be approximately 2.7% in gastric specimens and gastric biopsies performed during preoperative endoscopy, according to a recent review by Wang et al. [29]. In our study with a rat model, we found more fundic and antral foveolar hyperplasia in the SG group at 24 weeks postoperatively compared with the sham group. We also found antral intestinal metaplasia in one SG rat out of six (17%) in the context of reactive gastropathy. Three studies have previously described the histology of the gastric mucosa after SG in Wistar rats, with contrasting results. While Arapis et al. found no lesions in the fundic mucosa at 5 weeks postoperatively [20], Gulcicek et al. reported gastric lesions (foveolar hyperplasia, cystic glandular dilatation, and even fibrosis) at 4 weeks after SG [22]. Martin et al. also reported gastric foveolar hyperplasia and cystic dilatation of the glands in half of their rats at 4 weeks postoperatively and in all of them at 16 weeks postoperatively [21]. None of these three studies reported gastric intestinal metaplasia in rats after SG. These contrasting results in rats indicate that further studies should be performed to explore the pathological consequences of these findings and argue for long-term monitoring of the gastric mucosa after SG in rats, as in humans.

In humans, the presence of bile at endoscopy in the gastric sleeve has been reported by several authors. It was reported in 24% and 40% of endoscopies at 10 years, respectively, by Felsenreich et al. [14] and Csendes et al. [15], and even as high as 74% of endoscopies in the study by Genco et al. [12], explaining for the latter authors the high percentage of EB at 5 years in this study (19%). However, most authors have not described bile in the stomach at a pathological level at endoscopy after SG [10,11,13]. In our study, we found BA in the gastric lumen, but no significant difference was observed in the esogastric BA concentrations between SG and sham rats at 24 weeks postoperatively. This result could be due to a lack of power in the statistical analyses because of the small number of rats in each group. Of note, in the study by Siebert et al. [27], BA concentrations were higher in the gastric pouch of OAGB rats compared with sham rats at 30 weeks postoperatively, but they were not associated with esophageal lesions.

From these results, we may hypothesize that EHP appears to be non-pathological, as it is present in sham rats at the same frequency as in SG rats at 24 weeks postoperatively. In contrast, SG seems to cause gastric lesions, such as foveolar hyperplasia in both the antrum and fundus, without any evidence of a relationship with possible pathological bile reflux in our rat model.

Our study has several limitations. The first is the low number of rats, resulting in the low power of our statistical analyses for esophageal mucosa and bile acid analyses. However, the presence of antral intestinal metaplasia in one rat out of six (17%) at 24 postoperative weeks may alert us to the potential long-term consequences of SG on the gastric mucosa. Indeed, intestinal metaplasia is considered a pre-malignant lesion of gastric cancer in humans [30,31]. The second limitation is the absence of a reflux study by esophageal pH monitoring because this procedure is difficult to perform and not reproducible in rodents. Finally, like all studies of GERD in rats, the difference in esophageal anatomy (keratinized epithelium of quadruped animals), the dilatation of the SG (rarely observed in humans), and the unrestricted diet after SG (rats eat continuously, unlike humans) prevent us from extrapolating our results to humans.

## 5. Conclusions

Our results showing no more esophageal mucosa alterations 24 weeks after SG in male Wistar rats are reassuring compared with the results found in male Wistar rats after OAGB. However, gastric foveolar hyperplasia, both in the antrum and fundus, was more frequent 24 weeks after SG in our model, and this reactive gastropathy could be complicated by preneoplastic lesions, such as intestinal metaplasia. To study the occurrence of possible gastric lesions after SG in humans, it might therefore be wise to take advantage of the long-term follow-up recommendations for endoscopy after SG.

## Figures and Tables

**Figure 1 jcm-12-01848-f001:**
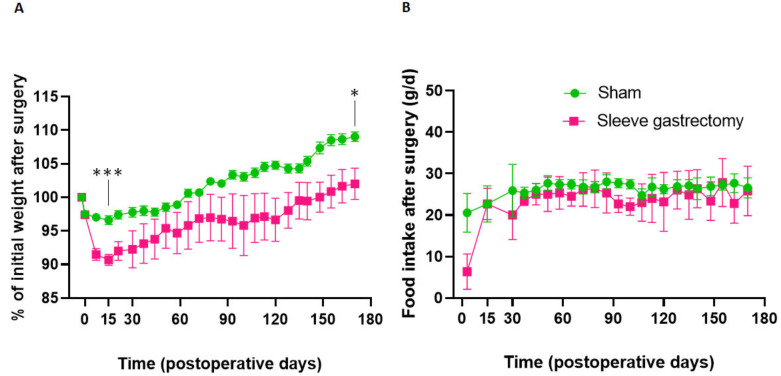
Body weight expressed as percent of preoperative weight (**A**) and daily food intake (**B**) in SG rats (n = 6) and sham rats (n = 8) during the 24 postoperative weeks. All values are expressed as means ± SEM, and comparisons between both groups were performed with Mann–Whitney tests after 2 weeks (maximal weight loss) and 24 weeks: * *p* < 0.5, *** *p* < 0.001.

**Figure 2 jcm-12-01848-f002:**
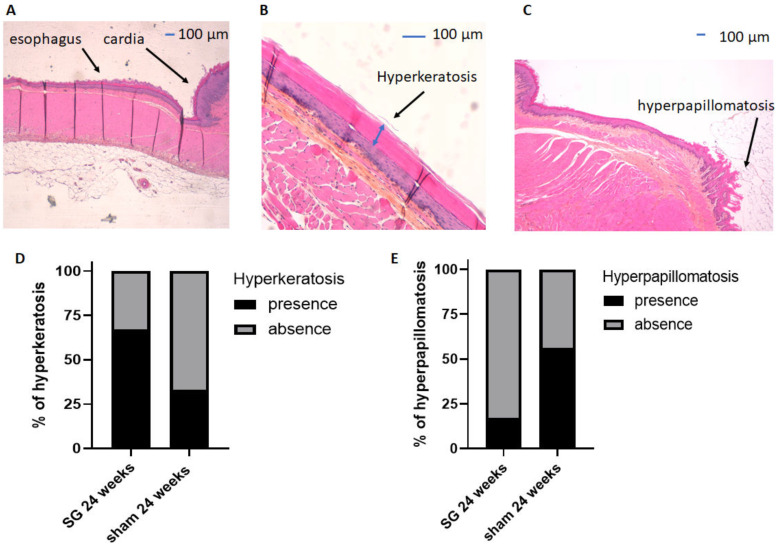
Analyses of HES staining of esophageal mucosa and comparison of the two groups of rats. Representative histology of healthy esophageal mucosa (**A**), esophageal hyperkeratosis (EHK) (**B**), and esophageal hyperpapillomatosis (EHP) (**C**). Comparison of the percentages of EHK (**D**) and EHP (**E**) after surgery between the two groups. SG: sleeve gastrectomy. All values are expressed as percentages, and comparisons were performed using Fisher’s exact tests.

**Figure 3 jcm-12-01848-f003:**
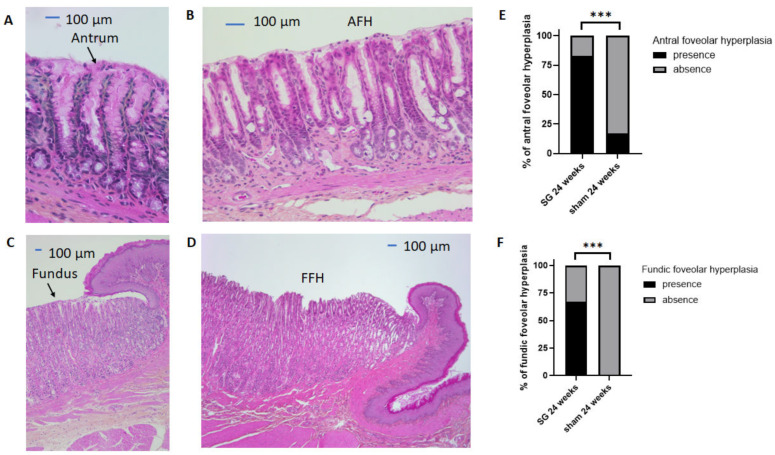
Analyses of HES staining of the antral and fundic mucosa and comparison between the two groups of rats. Representative histology of healthy antral mucosa (**A**), antral foveolar hyperplasia (AFH) (**B**), healthy fundic mucosa (**C**), and fundic foveolar hyperplasia (FFH) (**D**). Comparison of the percentage of AFH (**E**) or FFH (**F**) after surgery between the two groups. SG: sleeve gastrectomy. All values are expressed as percentages, and comparisons were performed using Fisher’s exact tests: *** *p* < 0.001.

**Figure 4 jcm-12-01848-f004:**
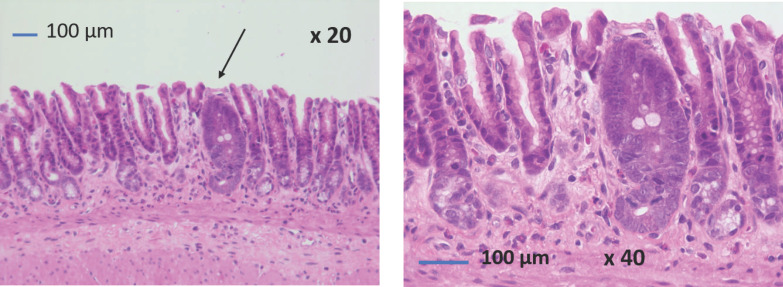
Antral intestinal metaplasia in one sleeve gastrectomy rat at 24 weeks postoperatively (×20 and ×40).

## Data Availability

The data presented in this study are available on request from the corresponding author.

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
