# Peer review of "No Long-Term Mucosal Lesions in the Esophagus but More Gastric Mucosal Lesions after Sleeve Gastrectomy in Obese Rats"

_jcm, 2023, doi:10.3390/jcm12051848_

Round 1

Reviewer 1 Report

Authors reported the study that showed gastric mucosal lesions after sleeve gastrectomy(SG) in obese rats, and concluded that long term endoscopic follow up was recommended in humans after SG  and useful to detect gastric lesions(page.8,line.249-251). 

Their experimental study on rats and the results which showed the gastric foveolar hyperplasia in SG rats at 24 weeks postoperatively were clear and significant.

However, in the discussion, authors referenced literatures about the gastric hyperplasia  in SG rats but not  about the finding in humans after SG. For the conclusion 'recommended in humans after SG  and useful to detect gastric lesions', they were recommended to reference other articles about human gastric lesion after SG and to discuss about the similarity between rats and humans. 

Author Response

Response to Reviewer 1

 Authors reported the study that showed gastric mucosal lesions after sleeve gastrectomy (SG) in obese rats, and concluded that long term endoscopic follow up was recommended in humans after SG  and useful to detect gastric lesions(page.8,line.249-251). 

Their experimental study on rats and the results which showed the gastric foveolar hyperplasia in SG rats at 24 weeks postoperatively were clear and significant.

However, in the discussion, authors referenced literatures about the gastric hyperplasia  in SG rats but not  about the finding in humans after SG. For the conclusion 'recommended in humans after SG  and useful to detect gastric lesions', they were recommended to reference other articles about human gastric lesion after SG and to discuss about the similarity between rats and humans. 

We thank the reviewer for her/his interest in our work.

As we mentioned in the Discussion (page 7, line 206), only one study reported findings of gastric biopsies after SG in 16 patients at 15 postoperative years [see reference 17]. The authors found “active gastritis” in 75% of the 16 patients, and no other type of gastric lesion was described [17]. Furthermore, the prevalence of gastric intestinal metaplasia in postoperative gastric biopsies has never been reported in humans, this point has been added in the Discussion (page 7, line 208). In contrast, the prevalence of preoperative gastric intestinal metaplasia is well known in humans, estimated to be approximately 2.7% in gastric specimens and gastric biopsies performed during preoperative endoscopy, according to a recent review by Wang et al. We have now added this point in the Discussion (page 7, line 210) and added the reference cited above [29].

We agree with the reviewer that extrapolation of these data to humans is not easy, which is why we have focused on this point in the limitations section (page 8, line 250).

Reviewer 2 Report

This research pushes the envelope towards building a stronger animal model to help us investigate long term sequelae of GERD in the setting of sleeve gastrectomy.

Both deaths were likely secondary to a gastric leak, rather than a fistula. a fistula denotes a chronic, epithelialized abnormal connection which would have not happened in the immediate postoperative period. would revise language. 

It seems there is initial weight loss, as expected from the sleeve gastrectomy, however there is regain up to preoperative weight by 180 days. In humans, when observed weight loss duration is not as pronounced or prolonged, one has to wonder if a significant portion of the fundus was retained during surgery (meaning not appropriately resected), meaning the sleeve was too loose, or improperly formed. Can the authors comment with slight more detail on the surgical procedure as is done on rats, and whether an endoluminar sizer (bougie type device) was used to create the sleeve. I suspect what authors have seen at 24 weeks as "residual stomach" "fundic surfaces" merely represents a sleeve with retained fundus. Would just accept that the possibility of a retained fundus is a limitation of an model.

GERD after sleeve gastrectomy is a controversial issue in that not all patients with reflux before a sleeve invariable worsen their disease (some improve) just as some without GERD before surgery go on and develop some. Ideally one would have to confirm that in fact, the obese rats actually developed evidence of GERD, which would then allow us to better understand these chronic effects. Also, gastric intestinal metaplasia is a known neoplastic lesion, however, these are oftentimes found incidentally BEFORE surgery, and do not represent reflux disease or an intrinsic result of altered anatomy (sleeve).

Round 2

Reviewer 1 Report

At the first review, reviewer recommended to 'reference other articles about human gastric lesion after SG and to discuss about the similarity between rats and humans'.

And authors responses were 'As  we mentioned in the Discussion (page 7, line 206), only one study reported findings of gastric biopsies after SG in 16 patients at 15 postoperative years [see reference 17]. The authors found “active gastritis” in 75% of the 16 patients, and no other type of gastric lesion was described [17]. Furthermore, the prevalence of gastric intestinal metaplasia in postoperative gastric biopsies has never been reported in humans, this point has been added in the Discussion (page 7, line 208).

And authors described ' extrapolation of these data to humans is not easy, which is why we have focused on this point in the limitations section, so that,the similarity between rats and humans were still not unclear. 

Therefore, it seemed hardly to be concluded that 'long term endoscopic follow up was recommended in humans after SG and useful to detect gastric lesions' using the data of the article. Authors were recommended to correct the expression of the conclusion. 
